# EF-UODA: Underwater Object Detection Based on Enhanced Feature

**Yunqin Zu [1], Lixun Zhang [1,\*], Siqi Li [2], Yuhe Fan [1] and Qijia Liu [1]**

[1] College of Mechanical and Electrical Engineering, Harbin Engineering University, Harbin 150001, China; zuyunqin@hrbeu.edu.cn (Y.Z.); fanyuhe@hrbeu.edu.cn (Y.F.); 1067555953@hrbeu.edu.cn (Q.L.)

[2] College of Shipbuilding Engineering, Harbin Engineering University, Harbin 150001, China; lisiqi_@hrbeu.edu.cn

[\*] Correspondence: zhanglixun@hrbeu.edu.cn

**Abstract:** The ability to detect underwater objects accurately is important in marine environmental engineering. Although many kinds of underwater object detection algorithms with relatively high accuracy have been proposed, they involve a large number of parameters and floating point operations (FLOPs), and often fail to yield satisfactory results in complex underwater environments. In light of the demand for an algorithm with the capability to extract high-quality features in complex underwater environments, we proposed a one-stage object detection algorithm called the enhanced feature-based underwater object detection algorithm (EF-UODA), which was based on the architecture of Next-ViT, the loss function of YOLOv8, and Ultralytics. First, we developed a highly efficient module for convolutions, called efficient multi-scale pointwise convolution (EMPC). Second, we proposed a feature pyramid architecture called the multipath fast fusion-feature pyramid network (M2F-FPN) based on different modes of feature fusion. Finally, we integrated the Next-ViT and the minimum point distance intersection over union loss functions in our proposed algorithm. Specifically, on the URPC2020 dataset, EF-UODA surpasses the state-of-the-art (SOTA) convolution-based object detection algorithm YOLOv8X by 2.9% mean average precision (mAP), and surpasses the SOTA ViT-based object detection algorithm real-time detection transformer (RT-DETR) by 2.1%. Meanwhile, it achieves the lowest FLOPs and parameters. The results of extensive experiments showed that EF-UODA had excellent feature extraction capability, and was adequately balanced in terms of the number of FLOPs and parameters.

**Keywords:** underwater object detection; feature extraction; feature fusion; YOLOv8

## 1. Introduction

Water bodies cover almost two-thirds of the Earth's surface, and produce almost half of the planet's oxygen while absorbing carbon dioxide from the environment [1]. We need to monitor critical underwater habitats in order to preserve underwater ecosystems. The abundance of natural resources in oceans [2] has also led to an increase in marine exploration activities. Noise is inevitably generated during the acquisition of visual information due to the complexity of the underwater environment and the difficulty of data acquisition, and poses a significant challenge to highly accurate underwater object detection.

Currently, one-stage object detection algorithms based on deep convolutional neural networks (DCNNs) have been shown to be effective in terms of object detection in recent years [3,4], including in underwater environments. Regression-based object detection algorithms, also known as one-stage object detection algorithms, have been proposed to solve the problem that object detection algorithms cannot realize real-time detection. Most state-of-the-art (SOTA) object detection algorithms are based on the single-shot multi-box detector (SSD) [5], the you only look once (YOLO) algorithm [6–11] and the fully convolutional one-stage object detection (FCOS) algorithm [12,13]. Yang et al. [14] proposed QueryDet based on RetinaNet [15] and FCOS, and designed a cascade sparse query mechanism. QueryDet reduces the computational cost of all feature-pyramid-based

object detectors and improves the detection of small objects by efficiently using high-resolution features of images while maintaining a high speed of inference. Hou et al. [16] proposed the idea that a deep neural network can learn the relationships between the given samples by itself, and used it to propose BatchFormer, which achieved satisfactory results in different experiments involving a small amount of available data.

Although the above algorithms have achieved suitably accurate results of detection in commonly encountered scenarios, some problems arise when directly applying deep-learning-based algorithms to detect objects underwater. First, acquiring underwater images is difficult because the scale of the captured image varies significantly with the distance between the camera and the object. Second, underwater images have poor quality, meaning the detection algorithm needs to have excellent feature extraction and generalization capabilities. Finally, although SOTA object detection algorithms are highly accurate, they require a large number of floating point operations (FLOPs) and parameters, which can make them difficult to train. Therefore, it is important to develop an underwater object detection algorithm with a robust feature extraction capability to meet the requirements of real-time detection and high accuracy while reducing the number of FLOPs and parameters used.

Following the design of one-stage detectors, here we proposed a one-stage object detection algorithm called the enhanced feature-based underwater object detection algorithm (EF-UODA). which could strike a suitable balance between detection accuracy and the number of FLOPs used, in order to satisfy the requirements of highly accurate real-time detection in complex underwater environments. The algorithm used an efficient feature extraction module and a sound approach to multi-scale feature fusion to highlight the feature-related information of underwater objects and strengthen its feature extraction capability. Moreover, it integrated advanced techniques, including the Next-ViT [17] and minimum point distance intersection over union (MPDIoU [18]) loss functions, to predict the classes and locations of objects of different sizes on four scales using multi-scale detection methods. Our main goal was to improve the accuracy of underwater object detection in real time by balancing the computational cost of the algorithm with its speed and accuracy of detection.

The contributions of this study can be summarized as follows:

(1)  We developed a convolution module called efficient multi-scale pointwise convolution (EMPC) for underwater object detection that outperformed traditional convolution modules while using fewer computations. We used it to design the C3-EMPC module, which had a better feature extraction capability than the original module.

(2)  We proposed a feature pyramid architecture called the multipath fast fusion-feature pyramid network (M2F-FPN). We implemented the multi-scale feature fusion aspect of the algorithm using two different connectivity modules and a new feature pyramid architecture. These methods effectively improved the feature extraction capability of the algorithm.

(3)  We presented EF-UODA, which was proved through extensive experiments to be a more accurate underwater object detection algorithm. The proposed method achieved real-time object detection, and it had high accuracy, and strong robustness.

## 2. Related Works

### 2.1. One-Stage Underwater Object Detection Algorithms

With advances in object detection technology and the exploitation of marine resources, underwater object detection has emerged as a popular direction of research in recent years. Several studies [19–22] have used vision-based object detection algorithms to explore underwater resources and marine badlands. Guo et al. [23] proposed an underwater object detection algorithm based on the improved multi-scale Retinex algorithm with color protection and YOLOv3 that solves the problems of blurring and low contrast of underwater images. Zeng et al. [24] introduced an adversarial occlusion network to Faster R-CNN so that the two networks compete with each other for learning. This prevents the detection network from overfitting to generate fixed features, and ensures high robustness

in the detection of underwater sources of food. Cai et al. [25] proposed collaborative weakly supervised detection by simultaneously using two detectors to learn clean samples from each other to improve the performance of the model. Chen et al. [26] proposed the sample-weighted hyper network (SWIPENET) and curriculum multi-class Adaboost to overcome ambiguity in underwater object detection in the presence of a large number of small objects by generating multiple high-resolution and semantically rich feature maps from the backbone network of SWIPENET. Wang et al. [27] proposed a paradigm for reinforcement learning to configure visual enhancement for object detection in underwater scenes, and experimentally verified its effectiveness. Zhang et al. [28] defined a new intersection over union loss based on YOLOv4 and achieved an accuracy improvement on the URPC dataset, but the GigaFLOPs (GFLOPs) were quite large and speed was not satisfactory. Chen et al. [29] proposed an underwater-YCC optimization algorithm based on YOLOv7 on the URPC dataset, which effectively improves the detection accuracy of small objects, but there was a lack of research on the drastic change of features in the URPC dataset. Guo et al. [30] proposed a lightweight underwater object detection algorithm based on YOLOv8s and achieved improvement on the URPC dataset, but there is a considerable discrepancy in accuracy compared to the SOTA algorithm.

Although the above methods are relatively accurate, they cannot satisfy the accuracy-related requirements of underwater object detection owing to subpar feature extraction capabilities and the fact that many algorithms use a large number of FLOPs and parameters. Designing a highly accurate detection method that has a robust feature extraction capability and uses a small number of FLOPs and parameters thus remains a major goal of research in this area. We develop such a method in this study.

### 2.2. Feature Extraction Module

The feature extraction module is an important part of a CNN, and is used to increase the depth and sensory field of the network to improve its feature extraction capability. Howard et al. [31] claimed that when a deep convolution was used to perform feature extraction operations on channels in each layer, the features obtained were aggregated on a single channel. This caused information on the features of individual channels to become independent, and enabled their point-by-point convolution. Singh et al. [32] designed heterogeneous convolutional filters where $3 \times 3$ convolution kernels and $1 \times 1$ convolution kernels were included in a single filter to extract features, and named it HetConv. Chen et al. [33] proposed octave convolution, which divided the convolution filter into a high-frequency component and a low-frequency component, processing the latter at low resolution to alleviate spatial redundancy, thus reducing the amount of computation while keeping the number of parameters the same. Zhang et al. [34] proposed SPConv, which divided the input channels into two groups for different types of processing. Han et al. [35] noted that the feature maps obtained with mainstream neural networks, used to enable a comprehensive understanding of the input data, inevitably contained redundant information. Qiu et al. [36] proposed SilmConv to reduce feature redundancy in the convolution process by reducing the number of feature channels and flipping weights.

However, traditional feature extraction modules have no adequate means of balancing the accuracy and computational cost of the algorithm. In this paper, we aim to reduce the number of parameters and the amount of required computations compared with traditional feature extraction modules while improving the accuracy of the algorithm.

### 2.3. Feature Pyramid Network

Effectively handling the multi-scale features extracted with neural networks is one of the main difficulties in object detection. The object detectors proposed in early research, such as SSD, are based on a pyramid feature hierarchy extracted using the backbone network for direct prediction. Lin et al. [37] pioneered FPN, which is a top-down multi-scale approach to feature fusion from deep to shallow features by fusing feature maps of neighboring layers to reduce the semantic gap between them. Inspired by FPN, PANet [38]

includes an additional path aggregation network from deep to shallow features. M2Det [39] contains a U-shaped module for multi-scale feature fusion, while NAS-FPN [40] uses a neural-network-based approach to searches in order to design the topology of the feature network automatically. Bidirectional FPN (BiFPN; [41]) uses weights for simple and fast feature fusion. Yang et al. [42] proposed AFPN, a progressive FPN that supports direct interaction in nonadjacent levels in order to avoid large semantic gaps between them by fusing neighboring low-level features and progressively fusing them at higher levels.

Although BiFPN and AFPN have achieved promising results, both feature pyramid architectures impose stringent requirements on the GPU, and cannot be adequately trained by using a single GPU. BiFPN is stacked three times in the neck architecture of EfficientDet to improve its accuracy, and while AFPN employs multiple feature fusion operations, it only uses normal convolutional components. In this paper, we aim to optimize multi-scale feature fusion in a more efficient way.

## 3. Methods

### 3.1. Architecture of EF-UODA

The architecture of the YOLOv8 is illustrated in Figure 1, and the architecture of the proposed EF-UODA is illustrated in Figure 2. The YOLOv8 architecture is composed of three key components: CSPDarknet (backbone network), PAN-FPN (neck network), and decoupled-head (detection head), and based on the idea of anchor-free.

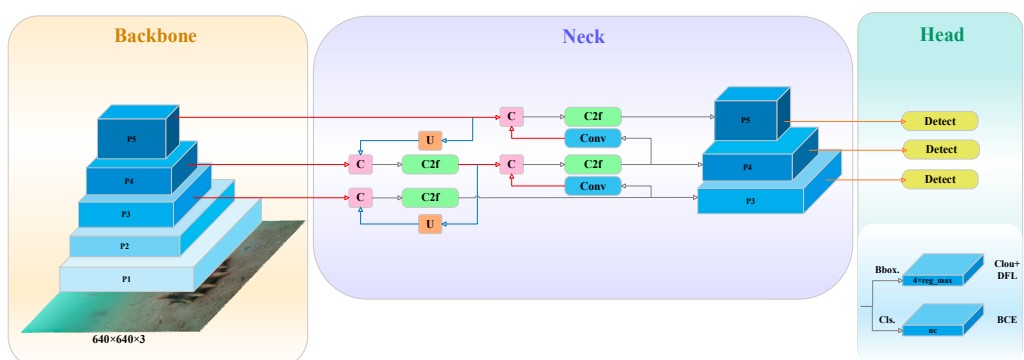

**Figure 1.** The architecture of YOLOv8.

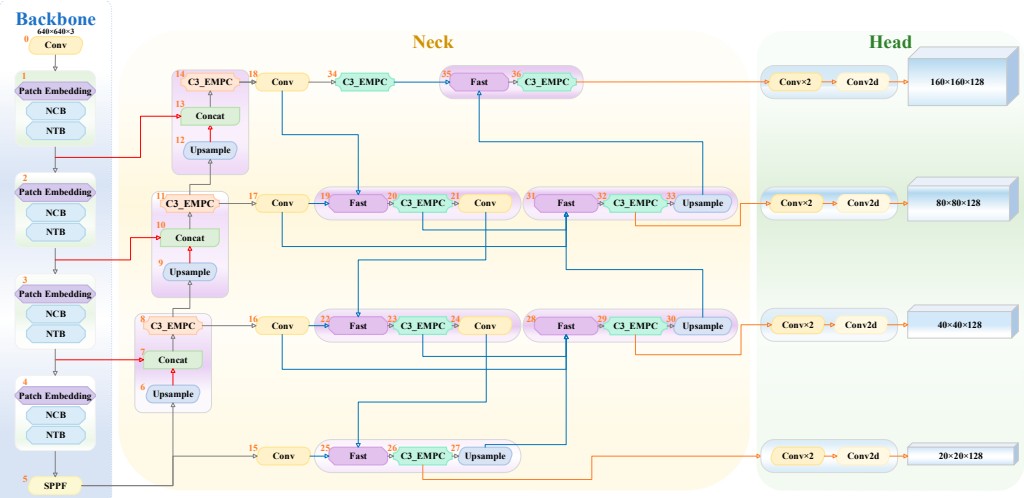

**Figure 2.** The architecture of EF-UODA. The backbone uses Next-ViT with an SPPF block at the end. The neck uses the M2F-FPN architecture. C3-EMPC: EMPC replaces the 3 × 3 Conv layer in the C3 bottleneck. The FAST block denotes fast normalized fusion.

We integrated Next-ViT to minimization of GFLOPs, which used a hierarchical pyramid architecture, into the backbone of EF-UODA using a patch embedding module, a next convolution block (NCB) module, and a next transformer block (NTB) module in one layer of the algorithm; collectively, these form the NexT layer in the proposed algorithm. When an image with a size of $640 \times 640 \times 3$ was imputed into it, it was reduced to $320 \times 320 \times 16$ after the first Conv (combined by Convolution, BatchNorm, and Sigmoid Linear Unit) layer. Following this, the scale of the feature image was reduced by half and the number of channels was doubled after each layer of NexT. To avoid the problem of image distortion caused by the scaling operation applied to the image as well as repeated feature extraction by the network. Our proposed M2F-FPN was used as a feature network in the neck to extract features at levels P2, P3, P4, and P5 of the backbone for one top-down and two bottom-up applications of multi-scale bidirectional feature fusion. These fused features were fed to the head of the network, which generated the predicted object class and the bounding box through one Conv layer and one convolutional layer. The head of EF-UODA was designed with a four-head architecture that could output scale-related information at four levels, P2–P5, to deal better with the effects of drastic changes in the scale of underwater objects.

### 3.2. Architecture of NexT

In the backbone of EF-UODA, we combine patch embedding, and Next-ViT's proposed NCB, NTB into one layer in the algorithm and named it the NexT layer, as described in Section 3.1. NCB and NTB were superimposed using the next hybrid strategy. Their architectures, presented in Figure 3, showed that multi-head convolutional attention (MHCA) was an important part of these two modules. MHCA was implemented by a multi-head convolution (group convolution) and a point-by-point convolution, as shown in Figure 3. MHCA took information regarding the input feature $z$ over $n$ channels and flattened it into $h$ parallel subspaces in the channel dimension. Based on the design of the multi-head paradigm, feature extraction was then carried out by using single-head CA, which are summarized by Equations (1) and (2):

$$\text{MHCA}(z) = \text{Concat}(\text{CA}_1(z_1), \text{CA}_2(z_2), \ldots, \text{CA}_n(z_n))W^P \tag{1}$$

$$\text{CA}(z) = \text{O}(W, (T_m, T_n)), where\ T_{\{m,n\}} \in z \tag{2}$$

where $W^P$ is a projection layer that facilitates the interaction of information between multiple heads. CA learns the affinities between markers in the local receptive field by optimizing the trainable parameters through iterations. $T_m$ and $T_n$ are neighboring features in the input feature $z$, $W$ is a trainable parameter, and $O$ is the inner product of the abovementioned parameters.

As shown in Figure 3, NCB followed the generalized architecture of MetaFormer [43], which consisted of MHCA and a fully connected layer (i.e., a multilayer perceptron (MLP)) based on residual concatenation. When feature vector $z^{in}$ was input into NCB the output of the MHCA layer was defined as $z^{td}$; $z^{td}$ and the output feature vector $z^{out}$ are expressed in Equations (3) and (4), respectively, as follows:

$$z^{td} = \text{MHCA}\left(z^{in}\right) + z^{in} \tag{3}$$

$$z^{out} = \text{MLP}\left(z^{td}\right) + z^{td} \tag{4}$$

NTB had an excellent capability to acquire global information. Figure 3 shows that it used the efficient multi-head self-attention (E-MHSA) module to capture low-frequency signals in conjunction with the ability of the MHCA module to capture high-frequency signals. The feature-related information of the input NTB was output in the form of mixed high and low-frequency information through collaboration between the E-MHSA module and the MHCA module. Finally, basic and unique feature-related information was extracted through the MLP layer.

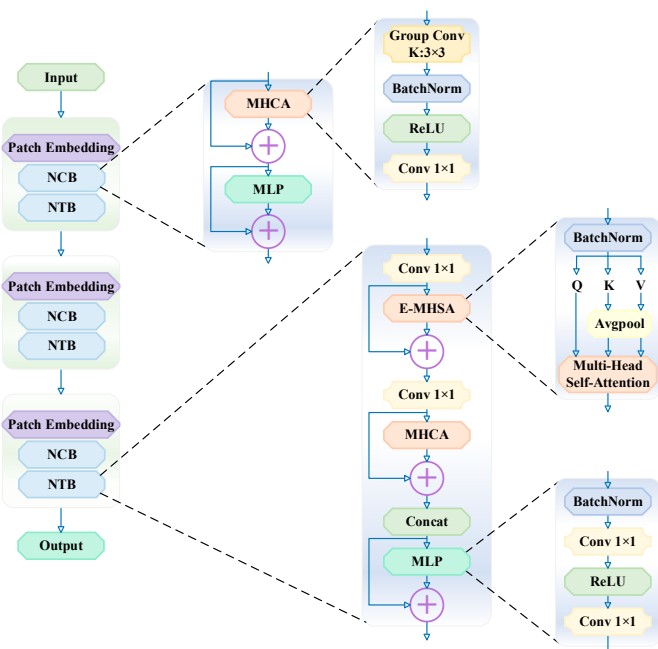

**Figure 3.** The architecture of the backbone.

The E-MHSA module is defined as shown in Equations (5) and (6):

$$\text{E-MHSA}(z) = \text{Concat}(SA_1(z_1), SA_2(z_2), \ldots, SA_n(z_n))W^P \tag{5}$$

$$\text{SA}(z) = \text{Attention}\left(XW^P, P_s\left(XW^K\right), P_s\left(XW^V\right)\right) \tag{6}$$

where $SA$ is a self-attention operator for spatial reduction; $W^P$, $W^K$, and $W^V$ are linear layers used for context encoding; and $P_S$ is the average pooling operation with step size $S$.

The output $z^{out}$ of the feature vector $z^{in}$ input to NTB can be expressed as follows:

$$z^{pw1} = \text{PointWise}\left(z^{in}\right) \tag{7}$$

$$z^{sa} = \text{E-MHSA}\left(z^{pw1}\right) + z^{pw1} \tag{8}$$

$$z^{pw2} = \text{PointWise}(z^{sa}) \tag{9}$$

$$z^{ca} = \text{MHCA}\left(z^{pw2}\right) + z^{pw2} \tag{10}$$

$$z^{cn} = \text{Concat}(z^{sa}, z^{ca}) \tag{11}$$

$$z^{out} = \text{MLP}(z^{cn}) + z^{cn} \tag{12}$$

where PointWise represents the point-by-point convolutional layer. The NCB and NTB modules based on the multi-head attention mechanism inherited the outstanding performance of the transformer module. Using them together not only enabled the current node with the ability to pay attention to the current pixel, but also to obtain the contextual semantics and, thus, local and global information. For complex underwater environments, the backbone combined with NexT and SPPF better coped with the problem of drastic changes in underwater object scale.

### 3.3. Efficient Multi-Scale Pointwise Convolution

The architecture of EMPC is shown in Figure 4. It was based on the ideas of the grouping operation and point-by-point convolution. Tensor $L$ with input $(w, h, c)$ was sliced into four groups by the rearrange function. The values of $w$ and $h$ of each new tensor were

unchanged, while the number of channels was changed to one-fourth of that of the original one. The convolution operation was performed on the four groups by using $1 \times 1$, $3 \times 3$, $5 \times 5$, and $7 \times 7$ convolution kernels, respectively, and the BatchNorm and Sigmoid Linear Unit activation functions were used to prevent overfitting. The feature-related information was merged into the channel dimension by the rearrange function to obtain a tensor $\dot{L}$ with dimensions $(w, h, c)$. Finally, information between different channels was exchanged by using a point-by-point convolution with one convolution kernel to realize the interaction between independent items of feature-related information in different channels and output the feature tensor $\widetilde{L}$ with dimensions $(w, h, c)$. The ratio, $r_p$, of the parameters required for one simultaneous feature extraction by the EMPC module and the convolution of a $3 \times 3$ convolution kernel was as follows:

$$r_p = \frac{3 \times 3 \times c \times c}{1 \times 1 \times \frac{c}{4} \times \frac{c}{4} + 3 \times 3 \times \frac{c}{4} \times \frac{c}{4} + 5 \times 5 \times \frac{c}{4} \times \frac{c}{4} + 7 \times 7 \times \frac{c}{4} \times \frac{c}{4} + 1 \times 1 \times c \times c} = \frac{36}{25} \qquad (13)$$

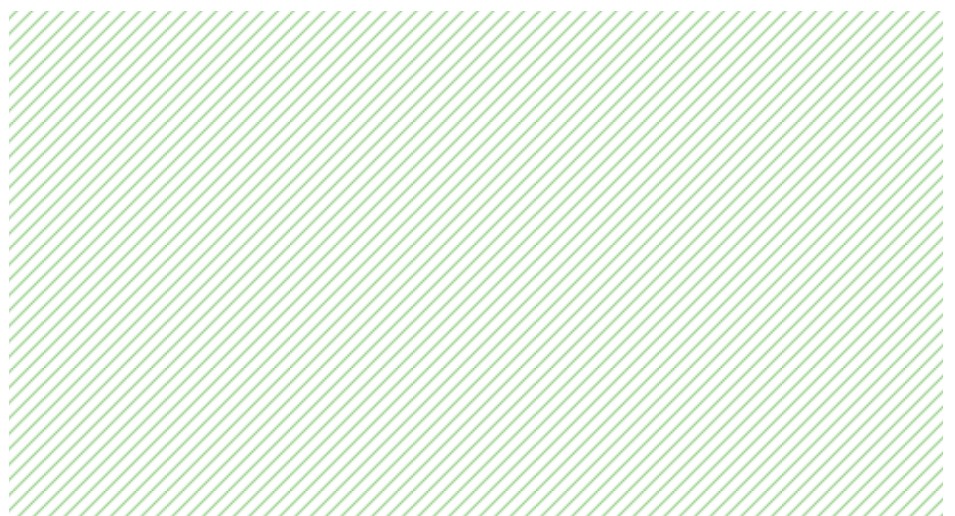

**Figure 4.** The combined architecture of C3-EMPC and EMPC.

The EMPC module had fewer parameters than the normal convolutional module, and reduced the amount of redundant feature-related information during feature extraction. The architecture of the C3-EMPC module is shown in Figure 4, in which the EMPC module was used to replace the $3 \times 3$ Conv layer in the bottleneck module [44] to upgrade dimensionality. This captured finer-grained feature information for situations where there was a significant disparity in the scale and features of the same underwater object, which in turn improved the accuracy of the algorithm.

*3.4. Architecture of M2F-FPN*

Different FPNs with input features at levels P2–5 are shown in Figure 5. The traditional top-down path-aggregated FPN is shown in Figure 5a, from which it was clear that the unidirectional flow of information in it limited the accuracy of its feature fusion. PANet added an additional bottom-up path-aggregated network to enhance the accuracy of feature fusion, and its architecture is shown in Figure 5b. Both FPN and PANet treated all the input features equally, without considering that different features had different resolutions and made unequal contributions to the output features. To solve this problem, BiFPN added an extra weight to each input feature to allow the network to learn different features differently. Its architecture is shown in Figure 5c. BiFPN was stacked three times in the EfficientDet network to improve the network's accuracy, where this increased the depth of the network and was not conducive to training with a CPU or a GPU with modest capabilities.

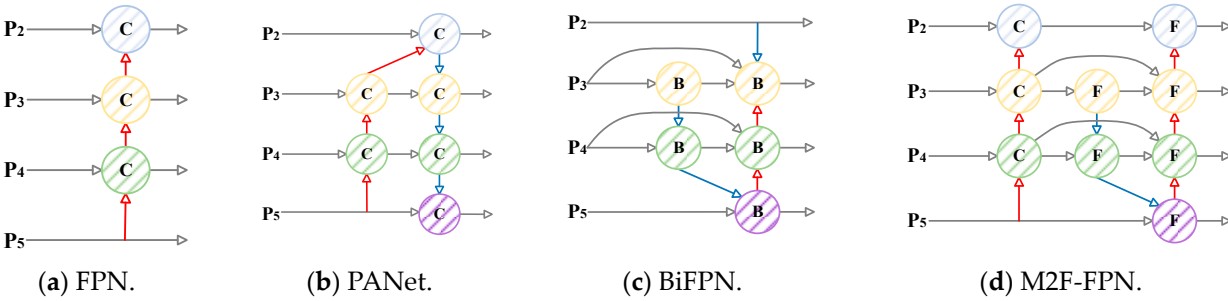

| (**a**) FPN. | (**b**) PANet. | (**c**) BiFPN. | (**d**) M2F-FPN. |

**Figure 5.** FPNs. (**a**) FPN introduces a top-down pathway to fuse multi-scale features from levels 2 to 4. (**b**) PANet adds an additional bottom-up pathway on top of FPN. (**c**) No duplicate BiFPN. (**d**) M2F-FPN, which has better accuracy/calculation trade-off.

PANet was more accurate than FPN and certain other FPNs, such as NAS-FPN, but requires more parameters and computation. We considered the architectures of PANet and BiFPN, and decided to remove nodes with only one input edge because they make only a small contribution to the feature fusion network. This simplified the architecture of the resultant network. When the original input and output nodes were in the same layer, an additional feature fusion channel was added to fuse more features without significantly increasing the computational cost. The architecture of M2F-FPN is shown in Figure 5d. We integrated the concat fusion module and the fast normalized fusion module into M2F-FPN. The results of our ablation experiments (Section 4) showed that this method of feature fusion delivered accurate results for datasets of underwater objects for detection. Concat fusion and fast normalized fusion are defined as follows:

$$\text{Fusion}_{cf} = \Sigma_i F_i \tag{14}$$

$$\text{Fusion}_{fnf} = \Sigma_i \frac{w_i}{\in + \Sigma_i w_i} F_i \tag{15}$$

where $w_i$ is an obtainable weight that is constrained to take a range of values by normalization, $F_i$ is the input to the fusion layer, and $\varepsilon = 0.0001$ is used to ensure numerical stability.

For example, when the resolution of the input image was $640 \times 640$, the feature level $P_3$ represented a feature map with a resolution of $80 \times 80$, and the output obtained by using the traditional FPN was represented as follows:

$$P_3^{out} = \text{Conv}\left(P_3^{in} + \text{Resize}\left(P_4^{out}\right)\right) \tag{16}$$

where Conv represented the convolution operation for feature processing and Resize represented the up-sampling or down-sampling operation to match the resolutions of features.

The output of M2F-FPN at feature level $P_3$, with the concat fusion and fast normalized fusion modules integrated into it, fused more feature-related information at different levels:

$$P_{3-1}^{out} = \text{Conv}\left(P_3^{in} + \text{Resize}\left(P_{4-1}^{out}\right)\right) \tag{17}$$

$$P_{3-2}^{out} = \text{Conv}\left(\frac{w_1 P_{3-1}^{out} + \text{Resize}\left(w_2 P_{2-1}^{out}\right)}{w_1 + w_2 + \in}\right) \tag{18}$$

$$P_3^{out} = \text{Conv}\left(\frac{w_1' P_{3-1}^{out} + w_2' P_{3-2}^{out} + \text{Resize}\left(w_3' P_4^{out}\right)}{w_1' + w_2' + w_3' + \in}\right) \tag{19}$$

where $P_{3-1}^{out}$ and $P_3^{out}$ represented the outputs of the first and second feature fusion layers at feature level $P_3$, respectively.

### 3.5. Head of the Object Detection Algorithm

The output of the resolution of the feature map of the head of our proposed EF-UODA at each scale (four scales in total) was represented by $S \times S \times C$. Here, $S \times S$ denotes the resolution of the output feature map and $C$ is its number of channels. Both concat fusion and fast normalization-based fusion were used in the neck of the algorithm. We fixed the number of channels, $C$, to 128 as this made it convenient to use the levy operation of each fusion module and helped implement the algorithm. Ge et al. [45] have noted that coupling the header impairs the performance and speed of convergence of an end-to-end algorithm. We thus used a decoupled header in the algorithm and followed the anchor-free design scheme.

We used the head of YOLOv8 to train the proposed algorithm. The bounding box complete intersection over union (CIoU) loss function used by YOLOv8 could not be optimized when the predicted box had the same aspect ratio as the actual labeled box but with completely different values for the width and height, and hence it did not cope well with the problem of drastically varying scales in underwater imagery. The MPDIoU function incorporated an existing loss function that considered all relevant factors. The idea of this bounding box similarity comparison metric based on MPD simplified the similarity comparability between two bounding boxes, which effectively coped with the problem of drastically different underwater image scales. Thus, we replaced the Bbox loss function with MPDIoU. Binary cross-entropy (BCE) loss was used as the loss function for classification, and the loss function of regression integrated the distribution focal loss (DFL) with MPDIoU loss, as shown in Figure 6.

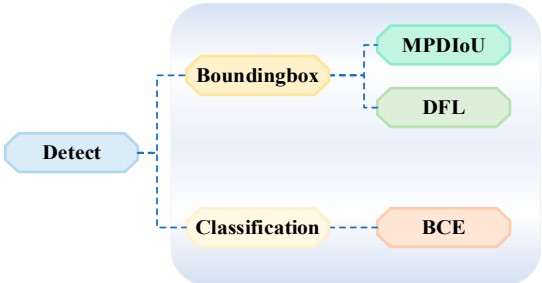

**Figure 6.** The architecture of the head.

The loss function of EF-UODA consisted of weighted forms of the above three loss functions. BCE, DFL, and MPDIoU are defined, respectively, as follows:

$$\text{BCE}(c_i, \hat{c}_i) = -c_i \log(\hat{c}_i) + (1 - c_i) \log(1 - \hat{c}_i) \tag{20}$$

$$\text{DFL}(\hat{s}_{i-1}, \hat{s}_{i+1}) = -((\hat{c}_{i+1} - c_i) \log(\hat{s}_{i-1}) + (c_i - \hat{c}_{i-1}) \log(\hat{s}_{i+1})) \tag{21}$$

$$\text{MPDIoU} = \text{IoU} - \frac{d_1^2}{h^2 + w^2} - \frac{d_2^2}{h^2 + w^2} \tag{22}$$

where $c_i$ and $\hat{c}_i$ denote the true and the predicted values of the model in the $i$-th cell, respectively, $\hat{c}_{i-1}$ and $\hat{c}_{i+1}$ are predicted values that are closest to the true value of $c_i$, and their probabilities are $\hat{s}_{i-1}$ and $\hat{s}_{i+1}$, respectively, where $\hat{s}_{i-1} = \frac{\hat{c}_{i+1} - c_i}{\hat{c}_{i+1} - \hat{c}_{i-1}}$ and $\hat{s}_{i+1} = \frac{c_i - \hat{c}_{i-1}}{\hat{c}_{i+1} - \hat{c}_{i-1}}$.

$h$ and $w$ are the height and width of the input image, respectively. $\text{IoU} = \frac{A \cap \hat{A}}{A \cup \hat{A}}$, where $A$ is the ground-truth box and $\hat{A}$ is the predicted box. $d_1^2 = (\hat{x}_1 - x_1) + (\hat{y}_1 - y_1)$ and $d_2^2 = (\hat{x}_2 - x_2) + (\hat{y}_2 - y_2)$, where $(x_1, y_1)$, and $(x_2, y_2)$ are the upper-left and lower-right corner points of the ground-truth box, while $(\hat{x}_1, \hat{y}_1)$ and $(\hat{x}_2, \hat{y}_2)$ are the upper-left and lower-right corner points of the predicted box, respectively.

## 4. Experiments and Results

### 4.1. Dataset

We reorganized the open-source URPC dataset created for the China Underwater Robotics Competition, and divided it at a ratio of 8:1:1 to generate the training set, validation set, and test set, respectively; thus, a total of 4434 images were obtained for training, 554 images for validation, and 555 images for testing. The URPC dataset was formulated by taking photographs of four categories of objects in an underwater environment, sea cucumbers, sea urchins, scallops, and starfish, as shown in Figure 7. The resolution of the images and the imbalance in the number of samples in each category constituted the major challenges posed by this dataset.

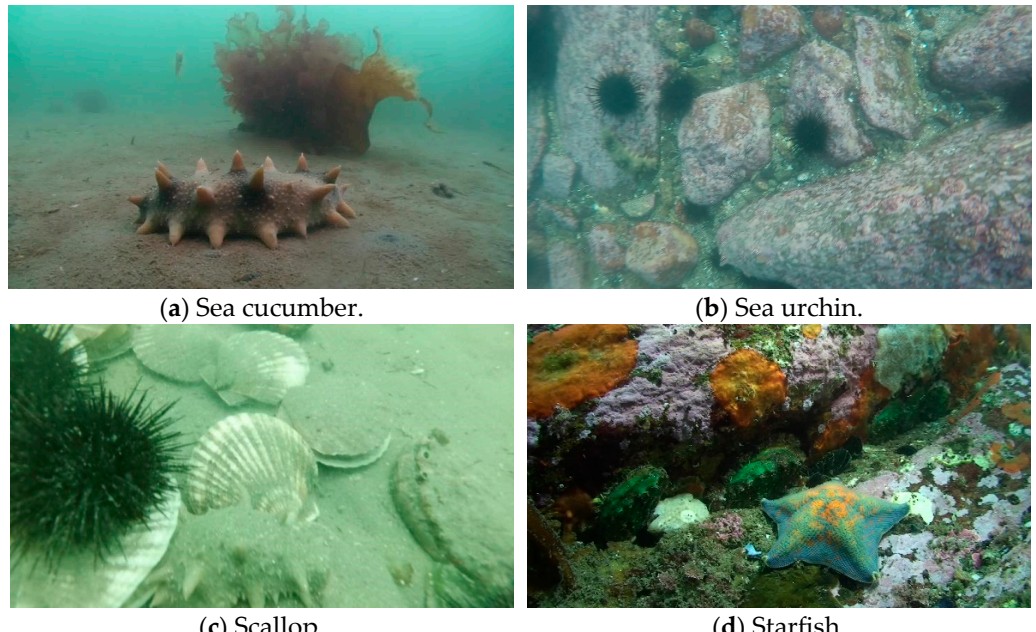

(**a**) Sea cucumber.        (**b**) Sea urchin.

(**c**) Scallop.        (**d**) Starfish.

**Figure 7.** Examples from the URPC dataset.

### 4.2. Experimental Equipment and Indicator of Evaluation

We used the Windows 11 operating system in our experiments. The CPU used was an i9-13900K, the GPU was an Nvidia GeForce RTX 4090 24G, and PyTorch2.0.0, CUDA12.1, and CUDNN8.9.0 were the deep learning frameworks. To ensure fairness in the experiments, the hyperparameters of each group were set to be the same in the training phase. Their settings are shown in Table 1.

**Table 1.** Hyperparameter settings.

| Training Epochs | Batch Size | Workers | Momentum | Weight Decay | Image Size |
|:---:|:---:|:---:|:---:|:---:|:---:|
| 100 | 4 | 8 | 0.937 | 0.0005 | $640 \times 640$ |

The loss function curves of the proposed method are demonstrated in Figure 8, which contain three parts: localization loss, distribution focal loss, and classification loss. As shown by panels (a) and (b) in Figure 8, the three loss functions for EF-UODA and YOLOv8X reached convergence within 100 training epochs. Therefore, the model converged in fewer than 100 epochs.

As the proposed deep neural algorithm was end to end, we compared it with a prevalent end-to-end deep neural algorithm in our experiments. As mentioned previously, we used the URPC dataset to evaluate our model, and reported its values of AP50 (i.e., the

mean average precision (mAP) at an IoU threshold of 0.5) under the COCO metric as the main indicator of evaluation.

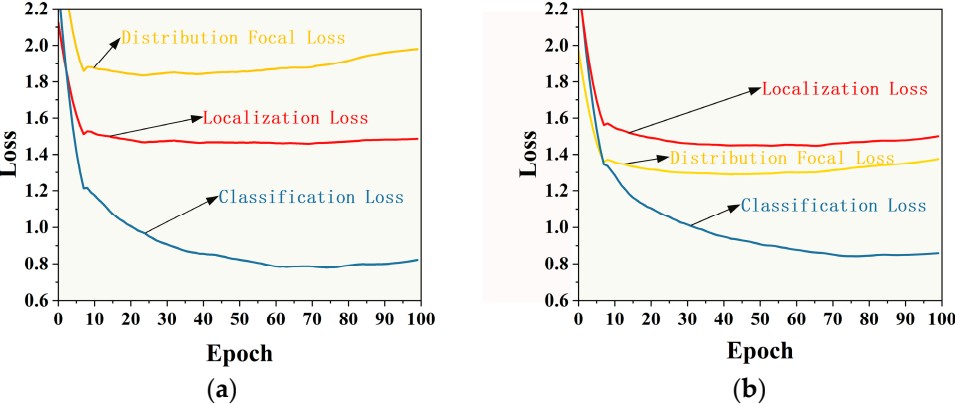

**Figure 8.** Curves of loss values after training for 100 epochs: (**a**) EF-UODA and (**b**) YOLOv8X.

The mAP calculated for objects in n categories was performed as follows:

$$\text{mAP} = \frac{1}{n} \sum_{i=1}^{n} \int_0^1 P(R) dR \tag{23}$$

$$P = \frac{TP}{TP + FP} \tag{24}$$

$$R = \frac{TP}{TP + FN} \tag{25}$$

where *TP* refers to the number of positive samples that are correctly detected (i.e., true positives), *FP* refers to the number of positive samples that are not correctly detected (i.e., false positives), and *FN* refers to the number of negative samples that are not correctly detected (i.e., false negatives).

### 4.3. Ablation and Comparison Experiments

For the exploration of underwater object detection algorithms, the main goal of this paper is to explore object detection algorithms with higher accuracy while maintaining real-time detection. The minimization of the algorithm GFLOPs and parameters will be explored when the above conditions are achieved. Therefore, under the premise of satisfying real-time detection, the accuracy of the algorithm is the most prioritized index to judge the performance of the algorithm.

To comprehensively assess the effectiveness of each scheme for EF-UODA, we conducted ablation experiments on the URPC dataset; the results are shown in Table 2. Replacing CSPDarknet with ViT-based backbone NexT effectively reduced the computational effort of the algorithm and made the algorithm more focused on contextual information, enhanced the generalization ability, and effectively ensured detection accuracy. Although the extra prediction head P2 and M2F-FPN increased the GFLOPs of the algorithm, the improvement in the accuracy of the algorithm is so significant (1.5% and 1.4% improvement in accuracy, respectively) that we believe it is worthwhile. From Table 2, it could be clearly seen that MPDIoU well addressed the problem of the vastly different scales of underwater images, which was very conducive to the URPC dataset, where the image scales change drastically.

We conducted ablation and comparison experiments to demonstrate the efficiency of the proposed feature extraction module C3-EMPC; the results are shown in Table 3. Additionally, the results of the experiments are depicted in Figure 9, facilitating the observation of data variations. Once the convolutional module of C3 was replaced with the EMPC module, the number of GigaFLOPs (GFLOPs) of the algorithm decreased from 142.3 to

140.0, the number of parameters used decreased from 34.81 M to 34.69 M, its frame per second (FPS) improved from 78.74 to 80.65, and its value of AP50 increased from 86.2% to 86.9%.

**Table 2.** Ablation experiment results.

| NexT | P2 | M2F-FPN | C3-EMPC | MPDIoU | GFLOPs | AP50 (%) |
|:---:|:---:|:---:|:---:|:---:|:---:|:---:|
| √ | | | | | 112.3 | 82.7 (+0.2) |
| √ | √ | | | | 137.6 | 84.2 (+1.5) |
| √ | √ | √ | | | 150.6 | 85.6 (+1.4) |
| √ | √ | √ | √ | | 140.0 | 86.1 (+0.5) |
| √ | √ | √ | √ | √ | 140.0 | 86.9 (+0.7) |

**Table 3.** Ablation experiment and comparison experiment results of the feature extraction module.

| Feature Extraction Module | GFLOPs | Parameters (M) | FPS | AP50 (%) |
|:---:|:---:|:---:|:---:|:---:|
| C3 | 142.3 | 34.81 | 78.74 | 86.2 |
| C3_CloAtt | 145.8 | 35.02 | 43.10 | 85.9 |
| C3_ScConv | 136.4 | 34.51 | 52.08 | 86.0 |
| C3_SCConv | 150.2 | 35.55 | 71.94 | 86.2 |
| C2f | 150.6 | 35.22 | 81.30 | 86.3 |
| C2f_Faster | 139.5 | 34.67 | 84.75 | 86.2 |
| C2f_DBB | 150.6 | 35.22 | 76.92 | 86.5 |
| C2f_ODConv | 135.7 | 35.28 | 57.80 | 86.0 |
| C3_EMPC | 140.0 | 34.69 | 80.65 | 86.9 |

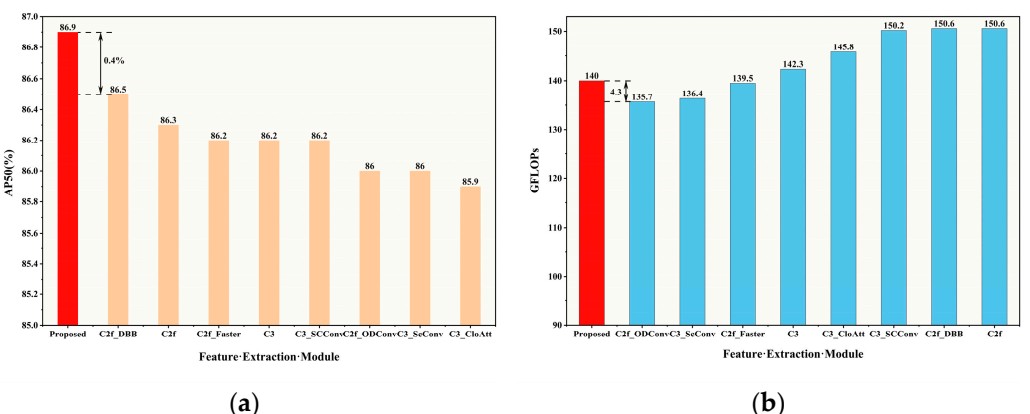

**Figure 9.** Comparison of feature extraction models. (**a**) Comparison of AP50 changes with different models. (**b**) Comparison of GFLOPs with different models.

We integrated a few high-performing methods into the C3 and C2f modules for the comparison experiments: C3-CloAtt [46], C3-ScConv [47], C3-SCConv [48], C2f-Faster [49], C2f-DBB [50], and C2f-ODConv [51]. C3-EMPC was slightly slower than C2f and C2f-Faster but its accuracy was almost 1% higher. The number of GFLOPs of C3-EMPC was slightly larger than those of C3-ScConv, C2f-Faster, and C2f-ODConv, but its speed of detection was also 1.55 and 1.40 times higher than those of C3-ScConv and C2f-ODConv, respectively, while its accuracy in terms of AP50 was 0.9% higher. The results of the ablation and comparison experiments in Table 3 demonstrate the efficiency of the C3-EMPC module. It provided the greatest improvement in the accuracy of the underwater object detection algorithms while better balancing the numbers of FLOPs and parameters as well as the FPS of the algorithm than the other modules.

To demonstrate the effectiveness of M2F-FPN, we compared it with the PANet and BiFPN architectures in experiments that used the same dataset and hyperparameters; the results are shown in Table 4. The AP50 value of the algorithm using M2F-FPN was higher by 0.3% and 1.2% than achieved with the PANet and BiFPN architectures, respectively. Although there was a slight increase in the number of FLOPs and parameters of the algorithm and a slight decrease in its speed of detection (FPS), we believed that this tradeoff was worthwhile in light of its higher accuracy. Our proposed M2F-FPN, which used both fast fusion and concat feature fusion, strengthened the multi-scale feature fusion of the overall algorithm and improved its accuracy.

**Table 4.** Comparison experiment results of feature pyramid network.

| FPN | GFLOPs | Parameters (M) | FPS | AP50 (%) |
|---|---|---|---|---|
| PANet | 130.9 | 34.34 | 82.64 | 86.6 |
| BiFPN | 132.4 | 34.49 | 81.33 | 85.7 |
| M2F-FPN | 140.0 | 34.69 | 80.65 | 86.9 |

We also conducted an ablation experiment to evaluate the contribution of the number of channels of the algorithm to its overall detection-related performance; the results are shown in Table 5. The EMPC module required dividing the number of input channels into four groups. We set the minimum number of input channels of the module to 64 to ensure the effectiveness of feature extraction. Given that a module was available to vary the scale of channels in the algorithm, the minimum number of channels was set to 128 to ensure the appropriate operation of the proposed deep neural algorithm. It was clear from the results in Table 5 that increasing the number of channels did not improve the accuracy of the algorithm. Its number of GFLOPs was 232.9 with 256 channels, and further increasing the number of channels was impractical when using a single GPU to train it. The algorithm had the smallest number of FLOPs and parameters with 128 channels, while the model recorded the highest accuracy of detection (AP50) of 86.9% on the test set while delivering the best generalization-related performance.

**Table 5.** Comparative experimental results using different amounts of channels.

| Channel | GFLOPs | Parameters (M) | AP50 (%) | Weight (MB) |
|---|---|---|---|---|
| 128 | 140.0 | 34.69 | 86.9 | 68.54 |
| 160 | 157.9 | 35.89 | 86.4 | 70.89 |
| 192 | 179.4 | 37.33 | 86.4 | 73.70 |
| 224 | 204.4 | 39.00 | 85.3 | 76.98 |
| 256 | 232.9 | 40.91 | 86.2 | 80.71 |

The loss function is an important component of deep neural algorithms that assess the predictions of the model. A well-defined loss function for the bounding box can significantly improve model performance. To determine the effectiveness of integrating MPDIoU as the loss function of the bounding box into the proposed EF-UODA, we compared its effect on the performance of our method with seven state-of-the-art loss functions on the same test set: CIoU, DIoU [52], GIoU [53], EIoU [54], SIoU [55], AlphaIoU [56], and Wise-IoU [57]. The experimental results are shown in Table 6 and Figure 10, from which it is clear that the algorithm delivered the best performance in terms of both accuracy and FPS when MPDIoU was used as the loss function of the bounding box. It recorded an AP50 of 86.9%, which was higher than that of GIoU (the second-best method) by 0.4%. Its FPS was 80.65, which was 1.15 times higher than that of DIoU (the second-best method). MPDIoU thus improved the FPS and accuracy of EF-UODA.

**Table 6.** Ablation experiment and comparison experiment results for the different bounding box loss functions.

| Bounding Box Loss Function | FPS | AP50 (%) |
|:---:|:---:|:---:|
| CIoU | 54.65 | 86.2 |
| DIoU | 69.93 | 86.0 |
| GIoU | 54.64 | 86.5 |
| EIoU | 68.49 | 86.3 |
| SIoU | 67.57 | 86.1 |
| AlphaIoU | 65.36 | 86.3 |
| Wise IoU | 61.35 | 86.1 |
| MPDIoU | 80.65 | 86.9 |

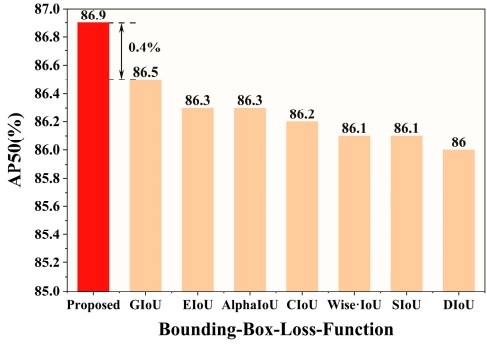

**Figure 10.** Comparison of AP50 changes with different bounding box loss function.

### 4.4. Comparison with Other Algorithms

We compared EF-UODA with SSD and Faster R-CNN on the same test set to verify its detection accuracy. SSD and Faster R-CNN use VGG16 and ResNet50 as their backbone, respectively, and the experimental results are shown in Table 7. Further, the trend of the speed indicator FPS is given in Table 7. Additionally, the results of the experiments are depicted in Figure 11, facilitating the observation of data variations. The mAP represents the average of all 10 IoU thresholds in the range [0.5:0.95]. EF-UODA recorded an accuracy that was 25.8% and 11.1% higher than those of the SSD and Faster R-CNN at an IoU of 0.5, respectively. Its accuracy in terms of the mAP metric was 20.4% and 8.0% higher than those SSD and Faster R-CNN, respectively. This demonstrated that our algorithm was significantly more accurate than the classical single-stage object detection algorithm SSD and the two-stage Faster R-CNN.

**Table 7.** Comparison experiment results on the URPC dataset.

| Algorithm | GFLOPs | Parameters (M) | FPS | AP50 (%) | mAP (%) |
|:---:|:---:|:---:|:---:|:---:|:---:|
| SSD | 353.7 | 26.2 | 23.62 | 61.1 | 32.4 |
| Faster R-CNN | 213.3 | 45.1 | 17.13 | 75.8 | 44.8 |
| RT-DETR | 247.1 | 74.67 | 104.17 (−9.29, +11.31) | 86.0 | 50.7 |
| YOLOv5X | 203.8 | 86.19 | 106.38 (−9.11, +10.99) | 83.0 | 48.8 |
| YOLOv7X | 188.0 | 70.80 | 57.47 (−2.69, +2.95) | 81.7 | 45.9 |
| YOLOv8X | 257.4 | 68.13 | 71.94 (−3.64, +4.04) | 82.5 | 49.9 |
| Proposed | 140.0 | 34.69 | 80.65 (−4.66, +5.26) | 86.9 | 52.8 |

We compared with the state-of-the-art ViT-based object detection RT-DETR [58] under the same conditions to illustrate the effectiveness of our proposed ViT-based EF-UODA; the results are shown in Table 7. RT-DETR used resnet-101 as its backbone. The accuracy of EF-UODA compared to RT-DETR in terms of AP50 and mAP was higher by 0.9% and 2.1%, respectively. EF-UODA's GFLOPs and parameters only accounted for 56.66% and 46.46%,

respectively, of those used by RT-DETR. Although the FPS of EF-UODA was 23.52 lower than that of RT-DETR, the FPS of EF-UODA achieved 80.69, which was sufficient for real-time detection. Considering that designing an underwater object detection algorithm with higher accuracy is the core purpose of this paper, and synthesizing the GFLOPs and parameters, EF-UODA is a better underwater object detection algorithm.

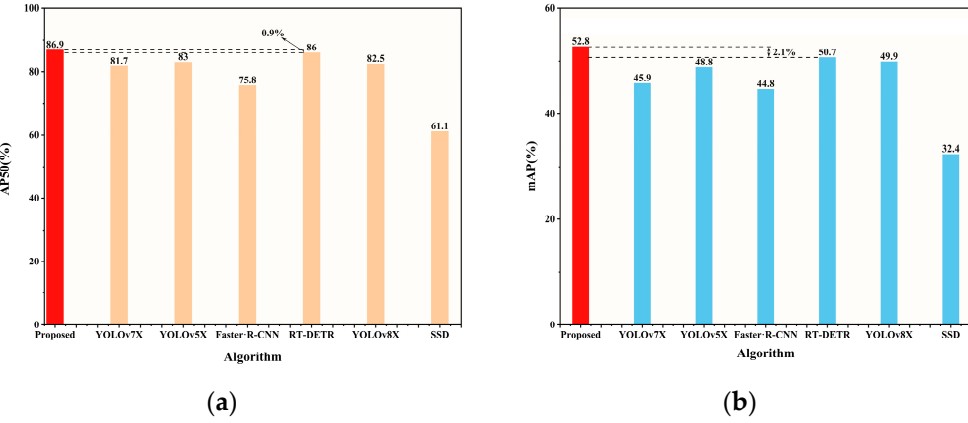

(**a**)  (**b**)

**Figure 11.** Comparison of different algorithms. (**a**) Comparison of AP50 changes with different algorithms. (**b**) Comparison of mAP with different models.

To demonstrate that our algorithm could accurately detect underwater objects, we compared it with SOTA one-stage object detection algorithms YOLOv5X, YOLOv7X, and YOLOv8X for the same test set. All four algorithms were based on the PyTorch framework, and the results are shown in Table 7. EF-UODA used 74.47% and 49% fewer GFLOPs and parameters than YOLOv7X, respectively, with an FPS that was 23.18 higher. Its accuracy in terms of AP50 and mAP was also higher by 5.2% and 6.9%, respectively. Moreover, EF-UODA used 68.69% and 40.25% fewer GFLOPs and parameters than YOLOv5X, and its accuracy in terms of AP50 and mAP was higher by 3.9% and 4.0%, respectively. We think that this increase in accuracy was worthwhile, despite the reduction of 25.73 in its FPS. Its FPS improved by 8.71 compared with the SOTA one-stage object detection algorithm YOLOv8X, while its accuracy in terms of AP50 and mAP was higher by 4.4% and 2.9%, respectively. It also used 54.39% and 50.92% fewer GFLOPs and parameters, respectively, than YOLOv8X.

We concluded that our proposed EF-UODA more accurately detected underwater objects than other SOTA object detection algorithms while reducing the number of FLOPs and parameters and ensuring a high speed of detection.

### 4.5. Comparison of the Detection Results

We randomly selected several images from the test set for experiments on underwater object detection by using YOLOv5X, YOLOv7X, YOLOv8X, and our proposed EF-UODA algorithm. The results are shown in Figure 12.

It is clear from Figure 12 that the dataset selected for the experiments consisted of fuzzy images, which made it difficult to detect objects in them because they were extracted from a real video captured underwater. This dataset, which contained complex backgrounds, thus imposed stringent requirements on the detection and feature extraction capabilities of the algorithm. YOLOv5X, YOLOv7X, and YOLOv8X omitted more information and made more incorrect predictions than the proposed method. In contrast, the EF-UODA algorithm exhibited better feature extraction capability through the C3-EMPC module, which was based on the idea of the grouping operation and the use of a multi-path fast fusion-based FPN for multi-scale feature fusion. It was thus able to detect underwater objects more accurately than the prevalent one-stage object detection algorithms YOLOv5X, YOLOv7X, and YOLOv8X.

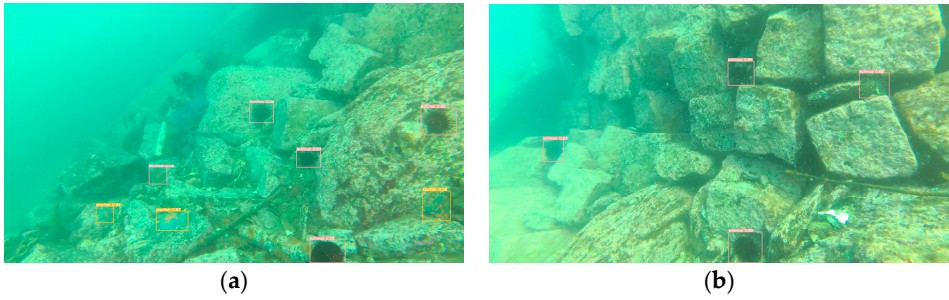

**Figure 12.** Comparison of detection results. (**a**) The ground truth. The detection results for (**b**) YOLOv5X, (**c**) YOLOv7X, (**d**) YOLOv8X, and (**e**) EF-UODA.

To prove the real effectiveness of the algorithm, we intercepted two images from the underwater video and reasoned through the proposed algorithm, and the results are shown in Figure 13. Figure 13a has five sea urchins and three sea stars, and Figure 13b has four sea urchins. It could be clearly seen that EF-UODA successfully reasoned out all the objects to be tested in the figure and gave a high confidence level, proving the effectiveness of the algorithm in the real world.

**Figure 13.** Real-world test results by EF-UODA. (**a**) and (**b**) images taken from real underwater videos.

## 5. Discussion

The combination of low-quality underwater images, drastic changes in scale, and high requirements for the generalization of algorithms has always been the difficulty of underwater object detection. The key to achieving excellent performance in EF-UODA is the efficient feature extraction and feature fusion capabilities, as well as the architecture that combines convolution with transformer. The ViT-based structure allows the algorithm to focus more on contextual information and effectively integrate high and low-frequency information. Moreover, the traditional loss function does not cope well with the problem of drastically varying scales in underwater imagery. MPDIoU based on the minimum point distance simplifies the similarity comparability between two bounding boxes, which can effectively cope with the problem of drastically different underwater image scales.

Compared to the improved YOLOv4 [28] proposed for the URPC dataset, the GFLOPs of EF-UODA are smaller and faster. In comparison to underwater-YCC [29] and lightweight YOLOv8s [30], our proposed algorithm focuses more on considering the high demand for underwater objects' drastic feature changes on the algorithm's generalization performance. Although the addition of the extra prediction head P2 and M2F-FPN made significant improvements in the accuracy of the algorithms, the increase in GFLOPs and parameters that they entailed made the deployment of the algorithms more demanding on the hardware of the embedded devices. In future work, more efficient methods can be explored to achieve further reduction in computation and storage costs while introducing extra prediction heads and M2F-FPN. In addition, we evaluated the algorithm only on the URPC dataset and continue to validate the performance of the algorithm on other underwater datasets in the future. Finally, we explore the introduction of the proposed improvements of EF-UODA into other advanced object detection algorithms in the future to further improve the performance of underwater object detection algorithms.

## 6. Conclusions

We proposed EF-UODA to solve the problems encountered in underwater object detection in complex underwater environments. It had strong feature extraction capability and high detection accuracy, and it could strike an appropriate balance between the amount of requisite computation and accuracy. The main contributions of this algorithm are summarized as follows. We developed convolutional modules with better feature extraction capability and proposed a feature pyramid structure with better feature fusion capability. Moreover, we integrated advanced technologies such as Next-ViT and MPDIoU into our proposed algorithm to form our state-of-the-art (SOTA) underwater object detection algorithm. Further, the effectiveness of the proposed method is proved by ablation and comparison experiments.

The results of our experiments showed that the proposed method recorded an accuracy in terms of AP50 that were higher by 3.9%, 5.2%, and 4.4%, respectively, than those of the SOTA one-stage object detection algorithms YOLOv5X, YOLOv7X, and YOLOv8X, and it had an mAP that was higher by 4.0%, 6.9%, and 2.9%, respectively, for the reorganized URPC2020 dataset. Meanwhile, EF-UODA surpassed SOTA ViT-based object algorithm by 0.9% AP50 and 2.1% mAP. Furthermore, it recorded a speed of detection of 80.65 FPS with 140.0 GFLOPs. In future work, we intend to deploy the EF-UODA in a remotely operated vehicle (ROV) to meet the high precision requirements of underwater grabbing, and we will further reduce the size of the model to enable embedded devices while maintaining its excellent accuracy and pursuing higher speeds. Moreover, we will conduct research on underwater image preprocessing to further improve the accuracy of the algorithm.

**Author Contributions:** Y.Z. conceived, planned, and performed the designs and drafted this paper. S.L., Y.F. and Q.L. provided guidance and reviewed this paper. L.Z. provided the design ideas and edited this paper. All authors contributed to the article and approved the submitted version. All authors have read and agreed to the published version of the manuscript.

**Funding:** This work is supported by the National Key Research and Development Program of China (Grant number 2020YFC2007700).

**Institutional Review Board Statement:** Not applicable.

**Informed Consent Statement:** Not applicable.

**Data Availability Statement:** Data are contained within the article.

**Conflicts of Interest:** The authors declare no conflicts of interest.

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
