# Peer review of "EF-UODA: Underwater Object Detection Based on Enhanced Feature"

_jmse, doi:10.3390/jmse12050729_

Round 1

Reviewer 1 Report

Comments and Suggestions for Authors

Authors presented a novel approach and neural network architecture for detection of objects underwater. Paper is well-structured, easy to read and used methods are explained in detail. Several experiments and test results are presented with results confirming that presented approach can be successfully compared to state-of-the-art.

There are some suggestions related to introduction and overview of references: - authors have focused only on ANNs while image preprocessing phase (using classical approach) for underwater images is omitted;

- adding link to github with used network model and code would be very useful to researchers. 

Author Response

Thank you very much for reviewing the manuscript (ID: jmse-2950743) entitled "EF-UODA: Underwater Object Detection Based on Enhanced Feature Extraction and Feature Fusion". The reviewers are acknowledged for their constructive comments on our manuscript. According to those helpful suggestions, we made a careful revision of the previous manuscript. In the revised manuscript, revised sections are highlighted with pink, purple, blue, light blue, red, green, orange, and dark red font colors, respectively. All revisions are explained as follows:

Reviewer #1:
Authors presented a novel approach and neural network architecture for detection of objects underwater. Paper is well-structured, easy to read and used methods are explained in detail. Several experiments and test results are presented with results confirming that presented approach can be successfully compared to state-of-the-art. There are some suggestions related to introduction and overview of references:

No.1: Authors have focused only on ANNs while image preprocessing phase (using classical approach) for underwater images is omitted;

Reply No.1: Many thanks to the reviewers for carefully reviewing the manuscript and providing constructive and valuable comments. In future work, we will conduct research on underwater image preprocessing to further improve the accuracy of the algorithm. In lines 568-570 of the revised manuscript.

No.2: Adding link to github with used network model and code would be very useful to researchers.

Reply No.2: Sincerely thanks to the reviewers for their valuable and thoughtful questions. We plan to participate in the Underwater Target Detection Contest this August and will provide a github link to the code after the contest is completed.

Once again, we thank the reviewer for the careful review and valuable comments. All of the revisions are highlighted in the word version through the review function. The pdf version is the latest version of the revised manuscript, in order to have a more complete reading experience, we did not show the revision process in the pdf version, but only showed the final revised results.

Reviewer 2 Report

Comments and Suggestions for Authors

Minor:

1. Many abbreviations are not explained at first mentioned, such as FLOPS, SOTA, mAP, GFLOPS, RT-DETR, etc.

2. Enhanced Feature Extraction- and Feature Fusion-based Underwater Object Detection Algorithm (EF-UODA) - it seems like EFE-FF-UODA? OR EF2F-UODA?

3.  Affiliation 1; 

 Affiliation 2;

It should be deleted.

4. [Error! Reference source not found.] - line 415

Major:

5. Figure 1: https://viso.ai/deep-learning/yolov8-guide/

It would be useful to compare figure one with the backbone from the link.

6. What is the reason for choosing YOLO v8?

7. The loss function of YOLO consists of three parts: classification loss, bounding box loss, and confidence loss. Which loss did you address in the paper? Or is it a combination of all the losses or various losses in various parts of the paper?

8. Regarding Table 3: Although there is a slight increase in AP50, there is a greater decrease to GFOPLS (more than 5%). You should define a cost function to provide the proof if this is acceptable in function of the goal. Maybe you should use some optimization algorithm to prove which algorithm is actually better when you take all parameters in count?

9. It is not clear what is the focus of your paper. You proposed MSF-FPN, but in the Abstract you mentioned EF-UODA. Hence, the Abstract does not match Section 4. It seems like you have 2 choices: to write 2 papers - one form UODA, one for MSF-FPN, OR to better integrate both into one picture (framework) and rewrite the paper as with integral approach. 

10. In comparisons with SOTA, you choose a lot of YOLO versions, which is not necessary. You should choose some other algorithms and only the latest YOLO. 

11. After proven on the dataset, you should add some real-word videos and see if the proposed function sufficiently in the real world. 

12. No sufficient references from 2023 and 2024 published in relevant (current contents) journals.

13. Finally, it is not clear, what would be your sensor platform: AUV, submarine, diver or diver's suit? 

Reviewer 3 Report

Comments and Suggestions for Authors

A new underwater object detection algorithm is proposed. It is based on improving feature extraction and feature fusion. Although the work is interesting, several aspects must be addressed before considering the publication of this work.

In the review of the state of the art, studies utilizing the URPC database should be analyzed and discussed, highlighting research opportunities.

Although contributions are listed at the end of Section 1 and different ablation experiment results are presented, the results of each contribution are not shown. That is, the final results are presented, and it is indicated that the advantages obtained are due to the combination of these contributions. However, this is not sufficient to demonstrate and justify each of the proposed improvements. Intermediate results should be shown that gradually demonstrate how each step or improvement you add improves the final results.

The text in some figures is not legible, for example, in Figure 1. It should be increased.

Are the parameters presented in Table 1 the best for your method? Are they optimized? For example, what happens if a different batch size is used? Results for other parameters should be added!

The results of Figure 8 are not clear. The images, at least, should be enlarged.

A discussion section should be added where: 1) the advantages and disadvantages of the proposed method are mentioned and 2) the different works reported in the literature are qualitatively and quantitatively compared.

For the results presented in Tables 2-6, some graphs should be added that allow observing maximum and minimum values, trends, averages, etc.

Section 5 should be called "Conclusions".

Comments on the Quality of English Language

 Minor editing of English language required

Reviewer 4 Report

Comments and Suggestions for Authors

Authors provided a manuscript about underwater objects recognition. In general the article is quite well organized, nevertheless I noticed some issues to be improved:

1. It should be no abbreviations in the title. Authors uses a lot of complex abbreviations that overloads the main text, I recommend to double check if  they all are really necessary.

2. Comparing results with methods from others research, authors could comment them a little bit instead of providing only references.

3. Figure 8 is too big and not informative, authors should search for better visualization solution that will highlight essential details.

4. Faulty reference in line 415.

Comments on the Quality of English Language

Quality of English is fine, minor editing required.

Round 2

Reviewer 2 Report

Comments and Suggestions for Authors

Regarding reply no2: Then , it should be EF (enhanced feature) UODA?

Regarding reply no 7: Figure is interesting. You should add explanation and /or this figure to the manuscript.

Regarding reply no 8: Tables 2 and 3: You missed the point. You haven't defined in the manuscript what does it mean "better algorithm". It is performed by  defined the goal as some sort of function that should be optimized or you should find extreme value of the defined function. From this, you can judge what is better.

Regarding replay 9: you should make it clear in the manuscript text.

Regarding reply 10: Table 7: The number are average values. It would be helpful to also see the rage: min-max.

Figure 11: This is redundant in this graph. It is not possible to connect the lines for different algorithms. You should find some other type of graph.

Regarding reply 11: You should add numerical results for these examples that it is clear about performance and the choice of the best algorithm.

Reviewer 3 Report

Comments and Suggestions for Authors

All the comments and suggestions have been addressed. This Reviewer recommends the manuscript's acceptance.

Author Response

We sincerely thank the reviewers for their suggestions on the manuscript, which greatly enhanced the quality of our manuscript. Finally, thanks again to the reviewers for giving acceptance advice on the manuscript.

Round 3

Reviewer 2 Report

Comments and Suggestions for Authors

The paper could be greatly improved by introducing minimization for GFLOPs.

Figures 9 and 10 should be arranged (the same type of graph) as figure 11 (which is reasonable).

Regarding reply to comment 10 (your replay 5): The algorithm do not provide constant fps, and also other metrics. It would be at least interesting to see minimal fps and maximal fps obtained by the experimental equipment.
